# Collapse of carbon nanotubes due to local high-pressure from van der Waals encapsulation

Cheng Hu [1,2,5], Jiajun Chen[1,2,5], Xianliang Zhou[1,2,5], Yufeng Xie [1,2,5], Xinyue Huang[1,2], Zhenghan Wu[1,2], Saiqun Ma[1,2], Zhichun Zhang[1,2], Kunqi Xu[1,2], Neng Wan[3], Yueheng Zhang[1,2], Qi Liang[1,2,4] & Zhiwen Shi [1,2,4] ✉

Van der Waals (vdW) assembly of low-dimensional materials has proven the capability of creating structures with on-demand properties. It is predicted that the vdW encapsulation can induce a local high-pressure of a few GPa, which will strongly modify the structure and property of trapped materials. Here, we report on the structural collapse of carbon nanotubes (CNTs) induced by the vdW encapsulation. By simply covering CNTs with a hexagonal boron nitride flake, most of the CNTs ($\approx$77%) convert from a tubular structure to a collapsed flat structure. Regardless of their original diameters, all the collapsed CNTs exhibit a uniform height of $\approx$0.7 nm, which is roughly the thickness of bilayer graphene. Such structural collapse is further confirmed by Raman spectroscopy, which shows a prominent broadening and blue shift in the Raman G-peak. The vdW encapsulation-induced collapse of CNTs is fully captured by molecular dynamics simulations of the local vdW pressure. Further near-field optical characterization reveals a metal-semiconductor transition in accompany with the CNT structural collapse. Our study provides not only a convenient approach to generate local high-pressure for fundamental research, but also a collapsed-CNT semiconductor for nanoelectronic applications.

High-pressure can strongly modify the structural, physical, and chemical characteristics of materials[1-4], thus has served as a powerful tool for the investigation in materials science[5-7], energy engineering[8-12], and biomedicine[13-15]. The high-pressures are typically produced in massive presses that focus large forces through two or more strong anvils to compress a sample[16-18]. The requirement of specialized equipment, however, has seriously limited the applications of high-pressure. One way to bypass this difficulty is to reduce the high-pressure time duration, i.e., the use of dynamic high-pressure[19,20], which can be easily achieved using a transient pulse loading. Another way, in principle, is to reduce the high-pressure region to a nanometer scale, so that a small force can generate very high-pressure, as pressure is inversely proportional to the loading area.

Van der Waals (vdW) heterostructure, assembly of low-dimensional materials through vdW force, has garnered significant attention in recent years due to the prospect of designing novel materials with on-demand properties[21-24]. One of the unique features of the vdW heterostructure is the nanoscale local high-pressure on the trapping materials, which is predicted to be on the GPa scale and a value of 1.2 GPa has been experimentally observed[25]. Such a convenient

[1]Key Laboratory of Artificial Structures and Quantum Control (Ministry of Education), School of Physics and Astronomy, Shanghai Jiao Tong University, Shanghai, China. [2]Collaborative Innovation Center of Advanced Microstructures, Nanjing, China. [3]Key laboratory of MEMS of Ministry of Education, School of Integrated Circuits, Southeast University, Nanjing, China. [4]Tsung-Dao Lee Institute, Shanghai Jiao Tong University, Shanghai, China. [5]These authors contributed equally: Cheng Hu, Jiajun Chen, Xianliang Zhou, Yufeng Xie. ✉e-mail: zwshi@sjtu.edu.cn

method of generating local high-pressure, through simply stacking of low-dimensional materials, is considered to create new chemical compounds and to explore the science of nanoconfined materials, such as the superconductivity in low-dimensional materials[26], phase transition[7,27,28], and hydrogen storage[29,30].

On the other hand, the collapse of CNTs has been discovered for a long time[31]. Various methods[32–38] have been reported for inducing the collapse of CNTs, such as applying hydrostatic pressure[34,35], using electron-beam irradiation[36,37] and employing ball milling[38]. Recently, Chen et al. have demonstrated the ability to collapse carbon nanotubes into nanoribbons less than 10 nm wide using high-pressure generated by diamond anvils and thermal treatment[4]. However, some methods are challenging to apply at the nano-micro scale, while others are complex and inefficient. Additionally, theoretical studies have presented a detailed collapse phase diagram of CNTs and identified the pressure range that CNTs with different diameters and walls could withstand without collapsing[39,40].

Here, we demonstrate that the local high-pressure generated by vdW encapsulation can induce a structural collapse of CNTs. By simply measuring the nanotube height, we found that structural collapse occurs in most hBN-encapsulated CNTs (≈77%). The collapse of CNTs is further confirmed by the Raman spectrum, where the G-peak has a prominent broadening from 11 cm$^{-1}$ to 63 cm$^{-1}$ with a blue shift of 7 cm$^{-1}$. The collapse of CNTs is fully captured by our molecular dynamics (MD) simulations, which reveal that the vdW encapsulation can create a local high-pressure of a few GPa at the nanometer scale. We further observe a metal-semiconductor transition in the collapsed CNTs through near-field optical characterization of their Luttinger liquid plasmons. Our approach provides a simple route to fabricate collapsed CNTs with tunable bandgaps and practical application in electronics and optoelectronics. Furthermore, the vdW encapsulation can serve as a simple way to generate local high-pressure of a few GPa for exploring high-pressure phenomena at the nanometer scale.

## Results

### Collapse of CNTs by hBN encapsulation

Figure 1a shows a schematic of the hBN-encapsulated-induced collapse of a CNT on a SiO$_2$ substrate. The single-walled CNTs were initially grown on a SiO$_2$ substrate through chemical vapor deposition, and then covered by an hBN flake through mechanical transfer. More details of the sample fabrication can be found in the "Methods" section. Due to the vdW attraction and the pinning effect between the rough SiO$_2$ surface and the hBN layer, a large tensile strain was established in the hBN flake when it encountered the CNT. This tensile strain in hBN results in a press onto the underneath CNT and leads to the collapse of the CNT.

In practice, we choose very thin hBN flakes with a typical thickness of 2 nm to encapsulate the CNTs. This is to avoid the thick hBN masking the following measurements of CNTs' height and Raman signal. Figure 1b shows the topography of a CNT with a diameter of 1.8 nm determined by atomic force microscopy (AFM). When the CNT was encapsulated by hBN, its height dropped from 1.8 nm to 0.7 nm (Fig. 1c), inferring a structural collapse of the CNT. The influence of structural bending of the hBN on the measurement of CNTs' height is analyzed in Supplementary Note 1, which can be neglected due to the small thickness of hBN. More data on this sample can be found in Supplementary Note 2. We have tested many CNT samples, and the height of all carbon nanotubes with and without the hBN encapsulation. As shown in Fig. 1d, the diameter of our CNTs without hBN coverage ranges from 1.0 nm to 2.0 nm. Interestingly, in the hBN-encapsulated area, most CNT heights drop to around 0.7 nm, corresponding to the structural collapse.

The collapse of CNTs is further confirmed by Raman spectroscopy. Figure 1e, f shows Raman spectra taken at two different positions of the CNT shown in Fig. 1b. Figure 1e shows the Raman signal

collected from circular CNT (purple dot in Fig. 1b), showing a sharp G peak with a full width at half maxima (FWHM) of only 11 cm$^{-1}$. The Raman spectrum taken from the collapsed CNT (blue dot in Fig. 1b) is shown in Fig. 1f, which displays several different features. Firstly, a Raman D peak appears around 1330 cm$^{-1}$, indicating the existence of structural imperfection or activation of a new scattering pathway[41]. The hBN phonon peak appears around 1360 cm$^{-1}$, which is from the encapsulated hBN layer. Secondly, the FWHM of the G peak increases from 11 cm$^{-1}$ to 63 cm$^{-1}$, indicating the diversity of the carbon-carbon bond resonant frequency, which provides clear evidence of structural collapse. Thirdly, the G peak also has a blueshift of about 7 cm$^{-1}$. The previous experiment[4] shows that the G-band has a blueshift at a rate of 5.4 cm$^{-1}$ GPa$^{-1}$ below 4.0 GPa. According to this, we can estimate the vdW pressure on the CNT by comparing the G-band blueshift to the CNT exposed outside the hBN. The blueshift 7.0 cm$^{-1}$ corresponds to an average pressure of 1.3 GPa. Using the convolution model, we can estimate the vdW pressure up to 10 GPa by Raman spectrum. More Raman data and details can be found in Supplementary Notes 3 and 4.

### MD simulations and theoretical analysis of the local high-pressure

To understand the occurrence of the high-pressure, we carry out MD simulations. Considering the tiny mismatch between hBN and carbon nanotube, which will lead to a rather large supercell, we replace hBN with graphene with the same lattice constant with CNT to save the calculation consumption. A more accurate fully atomistic MD simulation of CNT encapsulated by hBN with about 60,000 atoms is shown in Supplementary Fig. 6 and leads to the same result. Figure 2a shows a sectional view of the MD simulation of a graphene-encapsulated CNT.

Here, we set the CNT with a 1.8 nm diameter and fixed the bottom graphene. We add a tri-layer graphene on top of the CNT and apply a downward force on some atoms far away from the CNT in the top graphene. Once the upper layer of graphene approaches the lower layer of graphene, the graphene layers spontaneously stick together due to the vdW interaction, and eventually compress CNT into a collapsed phase and change the height of the structure. Figure 2b shows the interlayer vdW force and the pressure acting on the lower surface of carbon tubes. Noted that our MD simulations reveal a maximum local pressure of ≈10 GPa, which agrees well with previous predictions. The reason for generating such a high-pressure may be that the encapsulated 2D material film can effectively apply the downward vdW force from a larger area to a very localized CNT.

To ensure the contribution of the height change, CNTs with different diameters are simulated, as shown in Fig. 2c. The simulation result shows that whatever the diameter is, CNT will be severely compressed under the vdW interlayer interaction (Fig. 2c). The height fluctuation of the upper layer atoms is about 0.7 nm, which agrees with the AFM topography results (the dots). These simulation results show that even the 0.8 nm diameter CNTs can be also compressed into a complete collapse phase with a 0.34 nm interlayer distance. The critical diameter of the collapsed phase is reduced from 5.1 nm[42] to 0.8 nm caused by the unidirectional vdW pressure instead of the isotropic uniform pressures in Levy-Carrier equation[40], as shown in Supplementary Note 7. These simulation results, as well as extra simulations in Supplementary Note 9, shows that the collapse of CNTs by vdW pressure is theoretically possible.

### Statistics of the CNT collapse ratio

To estimate the yield of the collapsed CNTs through the hBN encapsulation, 84 as-grown CNTs and 15 hBN-encapsulated CNTs are counted for the statistics. Figure 3a, b shows the distribution histogram of the height of as-grown and of hBN-encapsulated CNTs on the SiO$_2$ substrate. Figure 3a shows the height distribution of the as-grown CNTs, which ranges from 1 nm to 2 nm. Figure 3b shows the distribution histogram of the height of hBN-encapsulated CNTs, where a

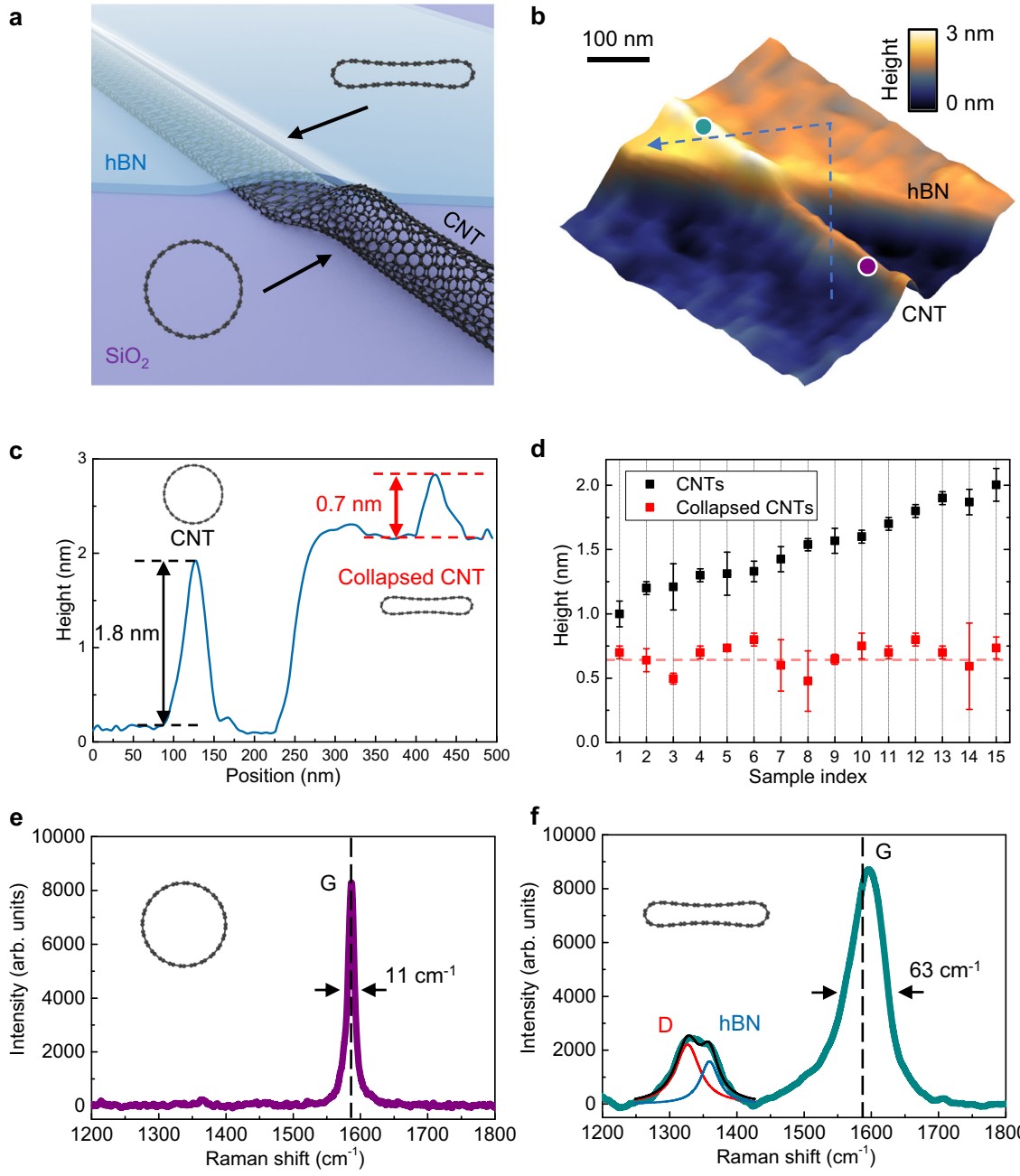

**Fig. 1 | Structural collapse of the hBN-encapsulated CNTs. a** Schematic of an hBN-encapsulated CNT squashing into collapsed CNT on SiO₂ substrate. **b** AFM topography of a hBN-encapsulated CNT with a diameter of 1.8 nm. **c** The topography line profile, shows a prominent change in height with h-BN encapsulation. **d** Statistics of the CNT height with (red squares) and without (black squares) hBN encapsulation. All collapsed CNTs, regardless of their initial diameters, exhibit a constant height of ≈0.7 nm. The red dashed line represents the theoretical height of the collapsed CNTs. The error bars represent the standard deviation of five height values measured at different positions of the same CNT. **e** Raman spectrum of a round CNT (purple curve) and **f** hBN-encapsulated collapsed CNT (green curve). The dashed lines indicate the initial G peak position of the round CNT. The red and blue lines are Lorentz fitting the D and hBN in-plane phonon peaks, respectively.

prominent reduction in the diameter is observed. Here, a height less than 0.9 nm is considered as the collapsed phase CNTs. The collapsed CNT yielding rate by the hBN encapsulation is around 77%, which is a rather high collapse ratio. Because of the uneven stress distribution in the hBN layer, a small portion of CNTs still survive. One possible method that can be used to improve the collapse ratio is to increase the thickness of the hBN layer. Our MD simulations reveal that a thicker hBN flake leads to a higher vdW pressure and a higher collapse ratio (see more details in Supplementary Notes 6 and 8). Additionally, encapsulation by other 2D materials, such as few-layer graphene, can also lead to the collapse of the CNTs (see more details in Supplementary Note 5).

## Collapse-induced metal-semiconductor transition

A metal-semiconductor transition is observed in accompany with the CNT's structural collapse. The metal-semiconductor transition of the CNT is experimentally revealed through near-field optical characterization of CNT's Luttinger liquid plasmons[43–46]. Here, we selected a specific CNT sample with both collapsed and circular parts in the hBN-covered region, so that we can directly achieve the collapse-induced change in its near-field optical response. Figure 4a shows the schematic of the near-field optical measurement of CNT plasmons under different gate voltages. An infrared laser beam of wavelength 10.6 μm is focused onto a gold-coated AFM tip, which can excite and probe the plasmons in the CNT simultaneously (more details are in the "Methods" section).

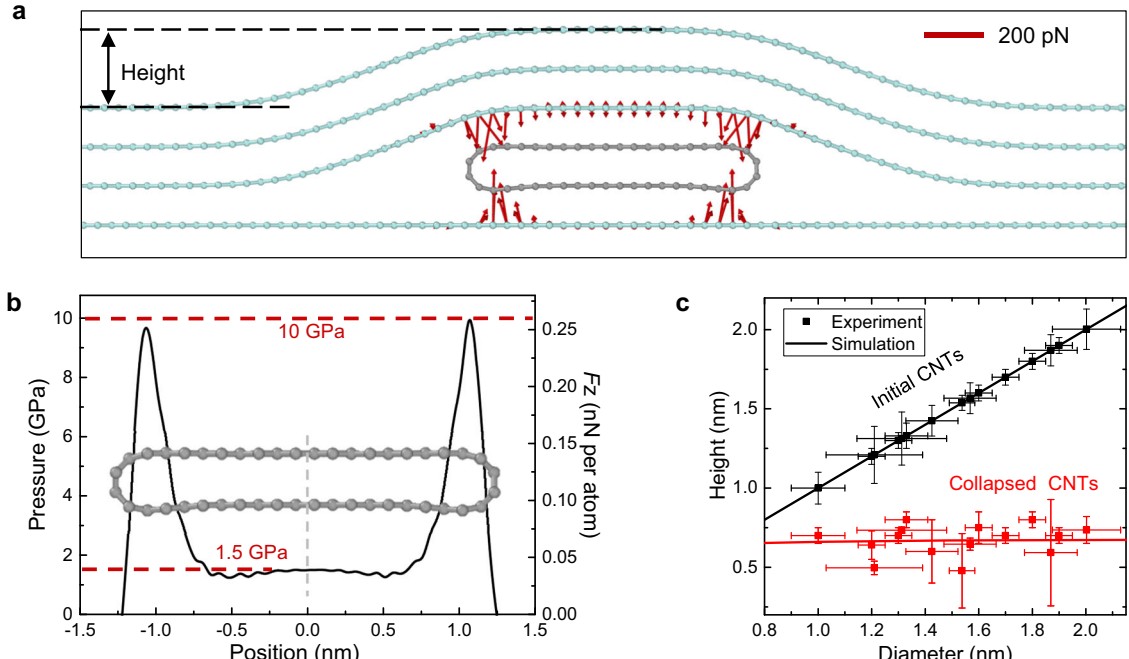

**Fig. 2 | MD simulations of the hBN-encapsulated CNTs. a** Sectional view of an hBN-encapsulated CNT simulated by MD. The red arrows represent the direction of the force on the atoms, and the lengths represent the magnitude. The length of the arrow in the upper right corner of the plot represents 200 pN. **b** Spatial distribution of the vdW pressure/force acting on the CNT. **c** Extracted height of the round CNT (black) and the hBN-encapsulated collapsed CNT (red). CNTs with diameters from 0.8 nm to 2.0 nm collapsed under the vdW pressure to around 0.7 nm. The error bars represent the standard deviation of multiple measurement values at different positions of the same carbon nanotube.

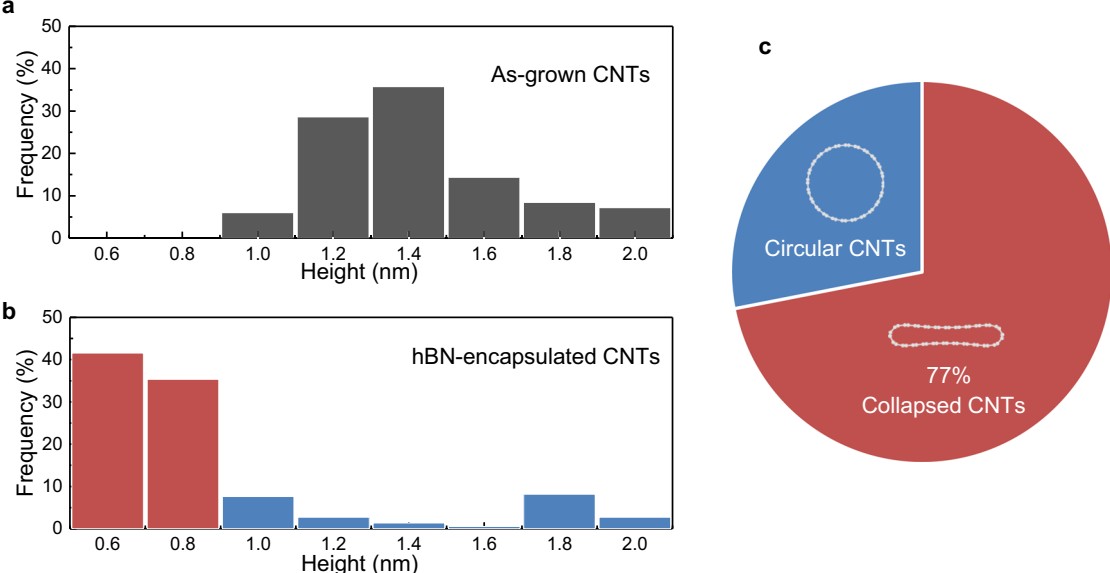

**Fig. 3 | Distribution histogram of CNTs' height before and after the vdW encapsulation. a**, **b** The distribution histogram of the CNTs' height before (**a**) and after (**b**) the hBN encapsulation, respectively. **c** The collapse ratio is around 77%. The percentage refers to the ratio of length between the collapsed CNTs and all CNTs investigated.

Previous studies have reported linear and non-linear Luttinger liquid plasmons in metallic and semiconducting CNTs, respectively. In metallic CNTs, the plasmon wavelength is constant, independent to the Fermi level controlled by the gate voltage. Whereas in semiconducting CNTs, the plasmon wavelength depended strongly on the Fermi level, and is tunable by the gate voltage (see more details for Luttinger-liquid theory of plasmons in CNTs in Supplementary Note 8).

Figure 4b shows the topography of the encapsulated CNT and the infrared response of plasmons in the encapsulated CNT under different gate voltages. Apparently, the collapsed region (upper part) has a much lower optical response than the circular region (lower part), especially at the charge neutral ($V_g = 0$ V). In addition, periodic patterns can be seen along the nanotube, which is a result of the plasmon interference. We extracted and plotted the plasmon line profiles for both the intrinsic and collapse regions in Fig. 4e, g, respectively. For the intrinsic circular region, the plasmon line profiles are of similar amplitude and period under different gate voltages (Fig. 4e). To quantitatively investigate the plasmon wavelength, we fit

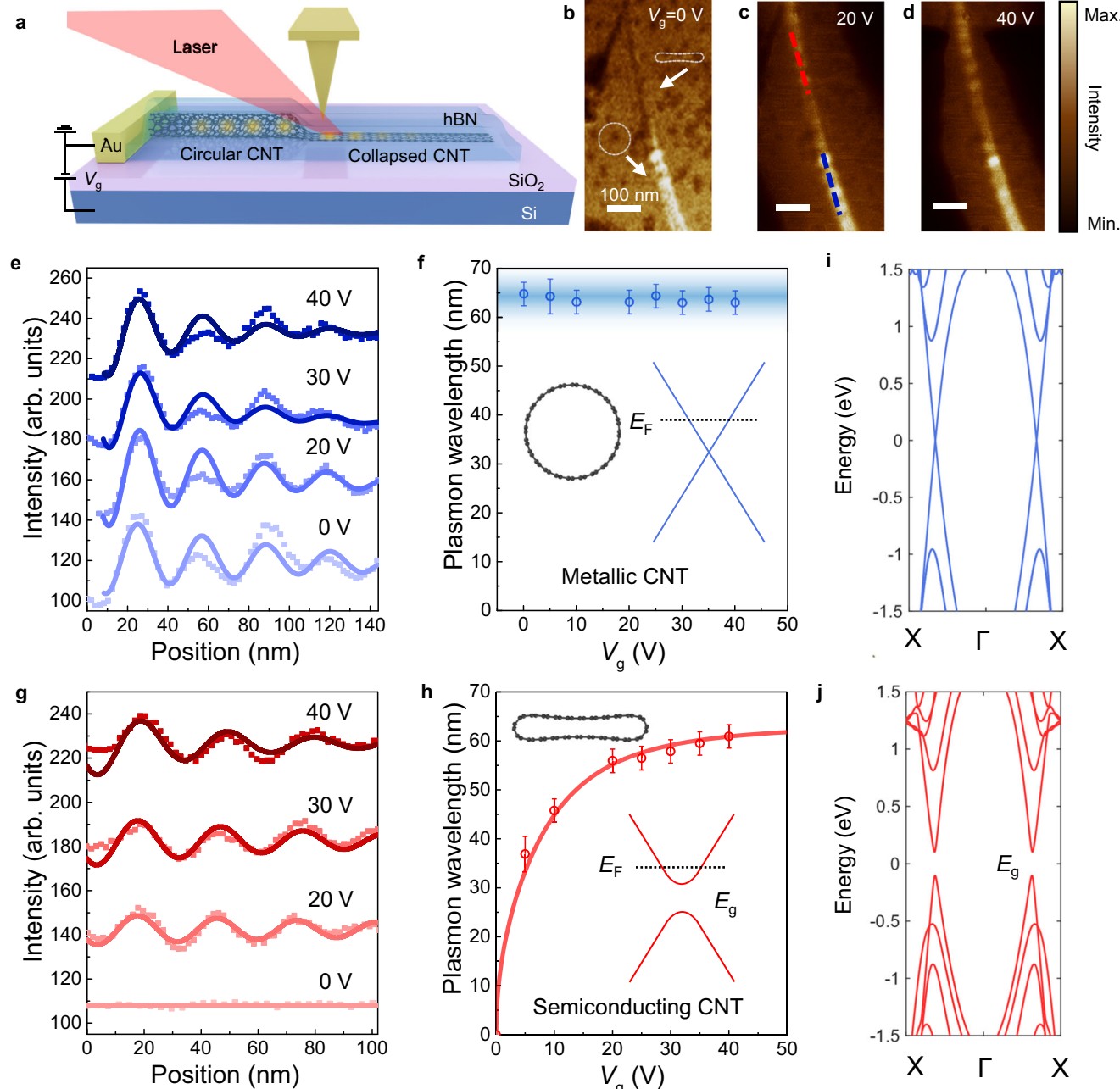

**Fig. 4 | Structural collapse-induced metal-semiconductor transition.**
**a** Illustration of the sample structure and schematic of the scanning near-field optical microscopy (SNOM). Part of the CNT collapsed due to the vdW pressure and formed a junction between the round and the collapsed CNT. **b**–**d** Near-field infrared images at different gate voltages of 0 V, 20 V, and 40 V. The scale bar is 100 nm. **e** The extracted plasmon profiles of the round metallic CNT (blue dots) and **g** of the collapsed part (red dots) under different gate voltages. Lines are fitting by the line profiles in Fig. 4e with an

exponentially decaying sine curves. **f** The extracted plasmon wavelength of the metal-type (blue dots) and **h** semiconductor-type one (red dots) with different gate voltages. The line is fitting by the theory of nonlinear Luttinger liquid plasmons in semiconducting CNTs. The error bars represent the standard deviation of multiple wavelength value measurements. **i** The calculated band structure of the intrinsic metallic CNT, and **j** the collapsed CNT, showing an opening of a bandgap.

the line profiles in Fig. 4e with an exponentially decaying sinewave function. The extracted plasmon wavelengths are plotted in Fig. 4f, where the plasmon wavelength keeps a constant value of ≈65 nm regardless of the gate voltages. Those plasmon features agree well with previously reported metallic CNT plasmons. Therefore, we identify the circular region to be a metallic CNT.

On the contrary, the plasmon line profiles in Fig. 4g for the collapsed region change dramatically with the gate voltage. At $V_g = 0$ V, there is almost no near-field optical response. With increasing the gate voltage, the near-field optical response as well as the plasmon

wavelength increases. The plasmon wavelength as a function of gate voltage is extracted and plotted in Fig. 4h, which exhibits a strongly nonlinear Luttinger liquid plasmon, in good agreement with previously reported plasmon behavior in semiconducting CNTs. Thus, we infer that the collapsed part turned out to be semiconducting. In addition to the example presented above, we have also observed metal-semiconductor transition in seven other collapsed CNT samples, as displayed in Supplementary Fig. 9.

To theoretically understand the observed metal-semiconductor transition, we carried out first-principles density functional theory

(DFT) calculations of the band structure of both collapsed and circular CNTs, as shown in Fig. 4i, j, respectively. When collapsed, the CNT's upper and lower walls get closer ≈0.3 nm with a dumbbell structure as shown in Fig. 2a. The interaction between the upper and the lower walls of the collapsed CNT changes its electronic band structure. As a result, an energy gap of ≈200 meV is opened, which can well explain the observed semiconducting plasmon behavior. In particular, the observed low optical response at $V_g = 0$ V is due to the lake of free charge carrier when the Fermi level is in the band gap of the semiconducting collapsed CNT. In addition, other transitions, such as from semiconductor to metal, are also predicted to occur in the structural collapse, which highly depends on the chirality of the CNT[2,47]. Correspondingly, the optical and electronic properties of the CNTs are structurally tunable simply through vdW encapsulation.

## Discussion

We have shown that the vdW encapsulation is able to generate a local high-pressure of a few GPa. This local high-pressure can lead to the collapse of CNTs with a collapsing ratio of up to ≈77%. The local high-pressure and collapse are fully captured by our MD simulations. We further show that the structure collapse of CNTs can open a bandgap in a metallic CNT and induce a metal-semiconductor transition. Our result provides a new method to fabricate collapsed CNTs with tunable energy bandgap, which is beneficial to their practical applications in electronics and optoelectronics. Furthermore, the vdW encapsulation can serve as a simple way to generate local high-pressure of a few GPa for exploring novel high-pressure phenomena at the nanometer scale.

## Methods
### Sample fabrication

Catalytic nanoparticles (Fe) were deposited on the $SiO_2$/Si chips through thermal evaporation (evaporation rate: $0.004$ nm s$^{-1}$, base vacuum pressure: $1 \times 10^{-6}$ mbar). Then the chips were put into a tube furnace (Anhui BEQ Equipment Technology), and gradually heated up to the CNT growth temperature (850 °C) under hydrogen and argon gas mixture at atmospheric pressure. When the growth temperature was reached, argon was replaced by methane to commence CNT growth. After a growth period of 60 min, the systems were cooled down to room temperature under a protective hydrogen and argon atmosphere[48,49]. After that, mechanically exfoliated hBN flakes on PPC film were transferred to the as-grown CNTs on $SiO_2$/Si chips. At last, the encapsulated CNT samples were exposed to hydrogen plasma at 280 °C to remove all organic residuals and contaminations. The obtained hBN-encapsulated nanotube density is very low, typically a few micrometers apart from each other, as shown in Supplementary Fig. 10. Standard electron beam lithography, electron beam deposition, and lift-off technology were conducted to locate the Au electrodes on the free-standing part of CNTs.

### Near-field IR imaging

A home-built scanning near-field optical microscope (SNOM) is used for the near-field infrared nanoimaging of plasmons. A $CO_2$ laser beam (10.6 μm) was focused onto the apex of a conductive AFM tip in the SNOM system. The enhanced optical field at the tip apex interacts with the CNT underneath the tip. The scattered light, carrying local optical information of the CNT sample, was collected by an MCT detector (KLD-0.1-J1, Kolmar) placed in the far field. Near-field optical images with spatial resolution better than 20 nm can be achieved with sharp AFM tips. Such near-field infrared images are recorded simultaneously with the AFM topography scanning during our measurements.

### Raman measurement

This measurement was carried out using a commercial Renishaw inVia Qontor confocal Raman microscope. A 532 nm wavelength laser beam

was focused onto the sample by a 100× objective lens (NA = 0.9). The power of the laser is about 2 mW. The grating groove density is chosen to be 1200 groove mm$^{-1}$.

### DFT calculations

The first-principles DFT calculations are performed using the Vienna ab initio simulation package[50], with projector augmented-wave method[51], and generalized gradient approximation of the Perdew–Burke–Ernzerhof functional[52]. A carbon bond length of 1.42 Å and a sufficiently thick (>10 Å) vacuum layer are used in the calculation. To simulate external vdW pressures applied on CNT by hBN in DFT calculations, we fix two hBN layers on both sides of the CNT and only relax the coordinates of carbon atoms until the forces are below $0.01$ eV Å$^{-1}$. Different fixed distances between two hBN layers correspond to different applied pressures on CNT.

### MD simulation

The simulated model system consists of a unit cell CNT (armchair, zigzag, and (22,1) CNT), a large graphene substrate, and a tri-layer top graphene. During the simulation, the substrate was kept fixed. The intra-layer interactions within the encapsulated CNTs model were computed via the AIREBO potential. The interlayer interactions between the CNTs and the graphene layers were described via the registry-dependent ILP with refined parametrization, which we implemented in LAMMPS. The initial configurations of the encapsulated CNTs were generated via geometry optimization using the cg algorithm with a downward force (2 eV Å$^{-1}$) added on the edge of the tri-layer graphene. The final configurations of the encapsulated CNTs were optimized with a threshold force value of $10^{-6}$ eV Å$^{-1}$.

### Reporting summary

Further information on research design is available in the Nature Portfolio Reporting Summary linked to this article.

## Data availability

The data that support the findings of this study are available from the corresponding author upon request.

## Code availability

The codes used in this study are available from the corresponding author upon request.

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

## Acknowledgements

This work was supported by the National Key R&D Program of China (No. 2021YFA1202902), the National Natural Science Foundation of China (Nos. 12374292 and 12074244), the open research fund of Songshan Lake Materials Laboratory (No. 2021SLABFK07), and the SJTU fund (No. 21X010200846). K.X. acknowledges support from the China Postdoctoral Science Foundation (No. 2022M712087).

## Author contributions

Z.S. and C.H. conceived this project. C.H. and Y.X. fabricated the device and performed the optical and electric experiments. J.C. carried out the MD simulation. X.Z. carried out the DFT calculation of the CNT band structure. N.W. provided the hBN crystals. C.H., J.C., X.Z., Y.X., X.H., Z.W., S.M., Z.Z., K.X., Y.Z., Q.L., and Z.S. discussed the results. C.H., Y.X., and Z.S. wrote the manuscript with input from all others.

## Competing interests

The authors declare no competing interests.
