## [Peer Review File · Nature Communications]

Collapse of carbon nanotubes due to local high pressure from van der Waals encapsulationEditorial Note: Parts of this Peer Review File have been redacted as indicated to remove third-party material where no permission to publish could be obtained.

REVIEWER COMMENTS

Reviewer #1 (Remarks to the Author):

Reviewer #2 (Remarks to the Author):

The manuscript titled "Collapse of Carbon Nanotubes by Local High-Pressure from Van der Waals Encapsulation" by Cheng Hu et al. presents a method for obtaining collapsed single-walled carbon nanotubes (SWCNTs) through the encapsulation of nanotubes under two-dimensional (2D) materials such as hBN or few-layer graphene. The authors claim that the pressure exerted on the SWCNTs reaches up to 10 GPa, leading to the structural collapse of CNTs. The experimental findings are supported by extensive AFM measurements and a few Raman measurements, complemented by MD and DFT calculations. Additionally, the authors report a metal-to-semiconductor transition in the collapsed CNTs through near-field optical characterization of Luttinger liquid plasmons in CNTs. The manuscript is well-written, and the presented findings have the potential to be of great interest to the scientific community. However, several claims provided by authors are not sufficiently backed up by the experiment. There are several significant issues that need attention before publication, as outlined below, and I recommend reconsideration of the manuscript following these revisions:

Major Revisions:

1. The novelty of the method for producing the collapsed CNTs is not sufficiently highlighted. Although the use of 2D materials to apply pressure to molecules was previously reported [Vasu et al., Nat. Comm. 7 12168 (2016)], it had not been applied to CNTs. I recommend referring to the review paper by He et al. [Small 2019, 15, 1804473], which outlines various methods of producing collapsed CNTs (see Section 3.3.4, "Collapse of CNTs by External Forces"), where the method presented by the authors is notably absent.
2. The provided Raman data appear limited, and in the reviewer's opinion, are insufficient to provide unequivocal evidence of the full structural collapse of CNTs. To address this, the authors should look for specific signatures of the collapsed CNTs in the Raman spectra, such as a significant increase in the intensity of the D-band and 2D-band (sometimes higher than the G-band), as reported in Figure 1e,f of Chen et al. [Nat. Electron. 4, 653–663 (2021)] and Figure 2 of Picheau et al. [ACS Nano 15, 596–603 (2021)] (not cited by authors although very relevant to the topic). It should be noted that the strong and sharp D-band is indicative of the activation of new scattering pathways due to cavities in collapsed CNTs and not the structural imperfection. Additionally, the disappearance of RBM modes should be observed due to the loss of cylindrical geometry, as reported in Chen et al. [Nat. Electron. 4, 653–663 (2021)]. These signatures are not clearly reported by the authors (except for the increase of the D-band intensity in Figure 1, however this fact was not discussed at all). Please provide Raman spectra for a statistically large number of CNTs from both the free-standing and collapsed regions (under hBN or graphene).
3. The observation (page 10, lines 202-203) that the same CNT can exhibit both circular and collapsed configurations under the same hBN flake is intriguing. The authors should provide further explanation and statistical data on the occurrence of such cases. Please explain why the collapse doesn't occur uniformly along length of the CNT. Raman mapping can be effectively employed to visualize how hBN encapsulation affects the Raman modes along the length of the CNT.
4. In general, it would be beneficial for the scientific community to understand how to distinguish between cylindrical and collapsed CNTs covered by hBN (or graphene) based on Raman spectra. Ideally, Raman maps of both free-standing and collapsed (under the hBN) regions of the same CNT should be measured to compare the frequency shifts and intensity variation in the D-band, G-band

and RBMs, as a function of coordinates on the sample surface.

5. Provide Raman spectra of pure hBN in areas free from CNTs to confirm that the peak at 1360 cm^{-1} originates from hBN and not from a D-band component of the CNTs. It would be instructive to see how the pressure estimated from the shift of the hBN line aligns with the estimated pressure from the CNTs' G-band shift. For reference, consider the frequency of few-layer (unstrained) hBN reported by Gorbachev et al. [Small 7, 465–468 (2011)].

6. The almost linear trend observed for the CNTs in Figure 1d is rather puzzling. It appears as if the CNTs were manually selected for presentation and do not follow the normal distribution of diameters, as shown in Figure 3a (top panel). Please clarify how the CNTs in Figure 1d relate to those in Figure 3a. If only few CNTs were selected for Figure 1d, please provide a separate figure with the full statistics, for instance in the SI. Furthermore, it is not clear how many CNTs were studied in total in Figure 1 and Figure 3. Please specify the exact number in the text, rather than just providing a percentage.

7. The authors repeatedly emphasize that they achieve local high pressures of up to 10 GPa. However, in the reviewer's opinion, their experimental data do not provide sufficient evidence to support such claims. The use of a convolution model, in this regard, is not a strong argument and raises further questions (see below). In fact, the average pressure estimated by the authors from the G-band shift of CNTs is only approximately 1.3 GPa. This estimation aligns closely with the findings of Vasu et al. [Nat. Commun. 2016], where a similar approach was employed to apply pressure to encapsulated molecules, resulting in a reported value of about 1.2 GPa. Furthermore, it appears that none of the works referenced in the review paper [He, Small 2019, 15, 1804473] report that the 10 GPa pressure is necessary for collapsing CNTs. Consequently, the assertion of reaching pressures of 10 GPa requires stronger justification and additional experimental data.

8. In SI Section S2, the discussion of the deconvolution results appears to yield strange results, particularly the authors should clarify how they interpret negative pressure shown in Figure S3 and its implications on CNT collapse. Overall this section looks rather weak in terms of arguments. Consider revising it.

9. The manuscript presents only one example of the metallic-to-semiconducting transition in collapsed CNTs, which is statistically insignificant, especially when claiming it as a new method for obtaining collapsed CNTs with a tunable energy bandgap. Please provide information about the number of collapsed CNTs that exhibit a semiconductor behavior. Again, Raman spectroscopy provides a perfect tool for studying metallic-to-semiconductor (or semiconductor-to-metallic) transitions since it should be reflected in the presence or absence of Breit-Wigner-Fano profile in the G-band spectrum (see Okuno et al. [PRL 111 216101 2013]).

10. In SI Section S3, it would be interesting to know if the authors measured Raman spectra for CNTs encapsulated in few-layer graphene and how the Raman lines of graphene and CNTs behave under strain. It will definitely have an added value for the community.

Minor Revisions:

1. On page 4, line 56, the references are incorrectly shown as "sample161920." Please correct the reference format.
2. Replace "fermi" with "Fermi" throughout the text (e.g., page 10, lines 211 and 213).
3. On page 13, line 266, change "Discussion" to "Conclusions."
4. Please increase the resolution of the figures in the SI. Some parts of the figures, such as the pie chart in Figure S4b, are hard to see when zoomed in.
5. Specify in the text how the error bars in Figure 1d (and other similar figures) were obtained.

Reviewer #3 (Remarks to the Author):

Reviewer #4 (Remarks to the Author):

The manuscript by Hu et al (NCOMMS-23-40743) represents an interesting study of sandwiching carbon nanotubes between hBN flakes (or few layer graphene) and a silicon oxide surface. The vdWs attraction between the hBN and the SiO₂ presumably generates enough clamping force normal to the surface to fully collapse the nanotubes. Metallic nanotubes are apparently rendered semiconducting. Modeling complements the experimental studies.

I find the topic quite interesting-- it applies the now-routine but powerful methods of 2-D vdWs heterostructure engineering to the mechanical and electronic physics of 1-D nanostructures. If the authors can successfully address my concerns stated below, the paper could yield a nice Nature Communications publication.

The paper is severely deficient in several respects:

1. The paper makes no mention of the rich history of collapsed carbon nanotubes, which dates back nearly 30 years. The reader is led to believe that this is the first experimental observation of collapsed nanotubes! Either the authors are woefully ignorant of the extensive literature on this topic, or they have omitted important references in order to falsely elevate the importance of their work.

The introduction should be rewritten to include important past work in this area, such as the original discovery of collapsed carbon nanotubes (N. Chopra et al, Nature 377, 135 (1995)) and studies of the mechanical stability of inflated/collapsed nanotubes (Barzegar et al, Nano Letters 16, 6787 (2016); Magnin et al, Carbon 178, 552 (2021); Xiao et al, Nanotechnology 18, 395703 (2007)).

2. The theoretical studies/simulations presented in the paper assume a perfect single wall carbon nanotube (e.g. a 10,10 tube). Yet the experimental part of the paper says almost nothing about the nanotubes used (other than that they were grown using CVD and what their apparent diameter is via AFM). CVD growth often yields highly defective tubes, with many kinks and "bamboo-like" internal structure. Such defects could completely alter the collapse energetics. How many walls do the nanotubes have? How defect free are they? If the tubes are multi wall, then the energetics are completely different, as only the inner-most walls contribute to vdWs energy lowering by collapse, whereas the outer walls only increase strain energy by collapse. Simply saying the thickness of the collapsed tubes as determined via AFM (through multiple layers of hBN or graphene) is that of bilayer graphene is not sufficient.

The authors must fully characterize the nanotubes employed, using high resolution TEM.

3. Several aspects of data presented in Fig. 1 don't make sense to me. In Fig. 1c, the FWHM of the topography profiles are identical for the inflated and collapsed tubes. How can this be? The hBN surely additionally smears out the line profile, and the collapsed tube is intrinsically wider than the inflated tube.

Also, for many (most) tube parameters, the schematic of Fig. 1a is unrealistic: The flattened portion of the tube significantly lowers the activation barrier or collapse, so the "exposed" part of the tube should spontaneously collapse as well. Do substrate interactions somehow keep it from zipping closed?

How are Raman spectra of Figs. 1e, 1f collected? Is the signal really from just one tube? Is some sort of near-field optical method used? I find it hard to believe that the Raman signal is collected from just the very small circular spots on a single tube as shown in Fig. 1b.

4. The paper suggests a metal to semiconductor transition induced by the hBN coverage and collapse.

But of course not all SWNTs are metallic to begin with. What happens if the tube is intrinsically semiconducting? Does a semiconducting to metal transition take place? As Lammert et al (Phys. Rev. Lett. 84, 2453 (2000), ref. 37 in the manuscript) show, tube collapse causes metallic (n,n) tubes to become semiconducting and can also cause small-gap semiconducting tubes to become metallic. Why is this not observed by the present authors?

5. The Supplementary Information has some confusing parts. For example, section S1 is entitled "The simulation of the influence of structural transition of the hBN", yet the text there (and Fig. S1?) deals with graphene, not hBN.

Reviewer #5 (Remarks to the Author):

In the manuscript titled "Collapse of carbon nanotubes by local high-pressure from van der Waals encapsulation" Hu, Shi and colleagues report an interesting phenomenon where 1D carbon nanotubes (CNTs) are collapsed by encapsulating them in between a 2D graphene crystal and SiO₂ surface. The main finding of the manuscript is that carbon nanotubes, regardless of their initial diameter, when placed in between a hBN flake and SiO₂ surface show a uniform collapsed thickness of 0.7 nm. They confirmed the collapsed structure mainly by AFM and Raman spectroscopy. Overall, the manuscript is well written and has very detailed theoretical simulations.

While I believe this is an important work and can have interesting applications as 1D metal semiconductor junctions, the manuscript lacks detailed characterisation and convincing experiments. My comments on the manuscript are presented categorically as follows:

1. The authors have stated that a high proportion (77%) of the CNTs flatten under the top hBN layers. Is this constant across all the samples measured? Since the deformation of the CNTs depend on the force applied on individual tubes, I would expect that if there is sufficient number of CNTs trapped in between hBN and SiO₂ layers, the tubes would avoid collapsing under the vdW pressure. Have the authors noticed any dependence on the number of CNTs trapped underneath the hBN layer and the percentage of collapsed tubes?
2. The authors should provide a larger area image of the encapsulated sample to give readers a better idea if there are bundles or individual nanotubes of encapsulated CNTs
3. I think the experimental section would benefit from more details of sample preparation especially how the hBN is placed on top of the CNTs. On a similar note, it is not mentioned if the Fe nanoparticle capping and the end encapsulation were removed before the transfer of hBN sheets.
4. Figure 1d raises quite a few questions. a) What is the red band in the figure stand for? b) do the sample index denote the number of nanotubes measured? In that case are there only 15 nanotubes trapped under the hBN flake? and c) how do the authors achieve the error bars? are they from measurements from the same nanotube at different positions or from different nanotubes of similar diameter?
5. In Figure 1e and f are the spectra collected from a single nanotube? Did the authors look at the RBM peak position of the collapsed and round nanotubes? It might give interesting information such as pressure and nanotube width. Moreover, the G-band broadening should also be analysed by deconvolution. The authors have carried out some deconvolution studies in figure S3- which is very difficult to understand and does not have any reference if any reported protocol was followed.
6. Have the authors considered the errors associated with AFM measurement of nanotube height as the nanotubes can deform under pressure from AFM tip itself? (Nanomaterials 2023, 13(3), 477)
7. The characterization of CNTs is incomplete. Are the CNTs synthesized single or multi walled? Is there any dependence of buckling pressure on the metallicity or chirality of the CNT? In their report R.S. Alencar et al. (Carbon, 2017, 125, 429-436) theoretically predicted that collapse pressure depends on the diameter of the nanotubes. Do the authors see any such dependence?
8. In Figure 2a, authors state that "The red arrows represent the direction of the force on the atoms, and the lengths represent the magnitude." The magnitude of force is impossible to understand without a scale bar.
9. As far as I understand, the authors only perform AFM measurements to characterise nanotube

dimensions. I do not understand how they have information for both height and diameter in figure 2c. Especially for collapsed nanotubes, I would expect their diameters to be higher compared to non-collapsed nanotubes. But that does not seem to be the case.

10. In figure 3, there is no information about how many nanotubes are considered for the statistical analysis. Moreover, larger nanotubes (1.8 nm and 2.0 nm) seem to be not as much affected by the vdW pressure. Is this assumption correct? If so, what are the possible reasons for such unusual behaviour?

11. The theoretical simulation for van der Waals pressure assumes a flat top and bottom surface. But experimentally, the authors have used SiO₂ as bottom substrate, which is not atomically flat. Would there be any difference in vdW pressure for an uneven substrate compared to a flat substrate?

12. Experimental details for scanning near-field optical microscopy sample preparation are lacking. I think the authors should add information about How the sample was prepared and the electrical contacts formed.

13. Since such semi-collapsed structure of CNTs would theoretically form 1D metal-semiconductor junctions, their electrical measurements should be very interesting. I would encourage the authors to include such kind of experiments in the manuscript.

Minor points:

1. The information about how the error bars are obtained and what they signify should be added to all the possible figures.

2. There is no Figure S2

3. In figure S4b, what does Length percentage signify?

4. Supplementary figures should be referenced as appropriate in the main text.

Remarks for referees

Response to Referee #1

Our reply: We thank the referee for sharing this information. It's great to hear about Nature Communications' initiative to support Early Career Researchers in peer review training. We appreciate the referee's dedication to fostering professional development in the research community.

Response to Referee #2

The manuscript titled "Collapse of Carbon Nanotubes by Local High-Pressure from Van der Waals Encapsulation" by Cheng Hu et al. presents a method for obtaining collapsed single-walled carbon nanotubes (SWCNTs) through the encapsulation of nanotubes under two-dimensional (2D) materials such as hBN or few-layer graphene. The authors claim that the pressure exerted on the SWCNTs reaches up to 10 GPa, leading to the structural collapse of CNTs. The experimental findings are supported by extensive AFM measurements and a few Raman measurements, complemented by MD and DFT calculations. Additionally, the authors report a metal-to-semiconductor transition in the collapsed CNTs through near-field optical characterization of Luttinger liquid plasmons in CNTs.

The manuscript is well-written, and the presented findings have the potential to be of great interest to the scientific community. However, several claims provided by authors are not sufficiently backed up by the experiment. There are several significant issues that need attention before publication, as outlined below, and I recommend reconsideration of the manuscript following these revisions:

Our reply: We thank the referee for his/her appreciation of our work and for their positive evaluation that our work is of great interest to the scientific community. We are also very grateful for the constructive comments provided by the referee that help us improve the manuscript.

1. The novelty of the method for producing the collapsed CNTs is not sufficiently highlighted. Although the use of 2D materials to apply pressure to molecules was previously reported [Vasu et al., Nat. Comm. 7 12168 (2016)], it had not been applied to CNTs. I recommend referring to the review paper by He et al. [Small 2019, 15, 1804473], which outlines various methods of producing collapsed CNTs (see Section 3.3.4, "Collapse of CNTs by External Forces"), where the method presented by the authors is notably absent.

Our reply: We appreciate the referee for his/her suggestion on emphasizing the novelty of our method. Following the referee's suggestion, we have added a new paragraph in the revised manuscript to review previously existing approaches for producing collapsed CNTs, which reads as:

"On the other hand, the collapse of CNTs has been discovered for a long time. Various methods have been reported for inducing the collapse of CNTs, such as applying hydrostatic pressure, using electron beam irradiation and employing ball milling. Recently, Chen et al. has demonstrated the ability to collapse carbon nanotubes into nanoribbons less than 10 nm wide using high pressure generated by diamond anvils and thermal treatment. However, some methods are challenging to apply at the nano-micro scale, while others are complex and inefficient. Additionally, theoretical studies have presented a detailed collapse phase diagram of CNTs and identified the pressure range that CNTs with different diameters and walls could withstand without collapsing."

Additionally, relevant references suggested by the referee are included in the revised manuscript.

2. The provided Raman data appear limited, and in the reviewer's opinion, are insufficient to provide unequivocal evidence of the full structural collapse of CNTs. To address this, the authors should look for specific signatures of the collapsed CNTs in the Raman spectra, such as a significant increase in the intensity of the D-band and 2D-band (sometimes higher than the G-band), as reported in Figure 1e,f of Chen et al. [Nat. Electron. 4, 653–663 (2021)] and Figure 2 of Picheau et al. [ACS Nano 15, 596–603 (2021)] (not cited by authors although very relevant to the topic). It should be noted that the strong and sharp D-band is indicative of the activation of new scattering pathways due to cavities in collapsed CNTs and not the structural imperfection. Additionally, the disappearance of RBM modes should be observed due to the loss of cylindrical geometry, as reported in Chen et al. [Nat. Electron. 4, 653–663 (2021)]. These signatures are not clearly reported by the authors (except for the increase of the D-band intensity in Figure 1, however this fact was not discussed at all). Please provide Raman spectra for a statistically large number of CNTs from both the free-standing and collapsed regions (under hBN or graphene).

Our reply: We thank the referee for his/her suggestion on providing more Raman evidence for the structural collapse of CNTs.

Following the referee's suggestion, we provide more Raman spectra taken from a large number of samples, as shown in Fig. R1a (from collapsed CNTs) and Fig. R1b (from free-standing CNTs). These sets of Raman spectra clearly demonstrate differences in the G- and D-peaks. Specifically, the G-peak in the collapsed-CNT Raman spectra appears significantly broader compared to the free-standing ones, as illustrated in Fig. R1c, with a full width at half maximum (FWHM) of around 60 cm^{-1} for the collapsed CNTs and around 25 cm^{-1} for the free-standing CNTs. This indicates a variation in the carbon-carbon bond resonant frequency due to the structural collapse. Additionally, the Raman D-peak is notably enhanced in the collapsed CNTs, potentially due to the activation of new scattering pathways resulting from cavities in the collapsed CNTs. These differences provide compelling evidence for the structural collapse of the CNTs.

Furthermore, we have included Raman mapping (Fig. R1 d-g) of a CNT that exhibits collapsed and free-standing structure, clearly demonstrating differences in D-peak intensity and G-peak width between the collapsed and free-standing parts. Fig. R1d shows the topography image of the sample, and Fig. e-g represent the near-field image, intensity of D peak mapping, FWHM of G peak mapping, respectively.

Regarding the 2D band of the CNTs, we noted that there is no definitive evidence for an increase in the 2D peak from the collapsed CNTs. Although Fig. 1f of Chen et al paper shows an increase of the 2D peak after structural collapse, the Fig. 5e of the same paper show a decrease of the 2D peak. In our experiments, we found that the 2D peak intensity decrease after collapse. Therefore, the change of the intensity of the 2D peak is still elusive, and we prefer not using the 2D peak as an indication of the structural collapse.

Regarding the RBM mode, as mentioned by the referee it should disappear due to the loss of cylindrical geometry. However, it is difficult for us to observe the RBM peak. First, the RBM peak is very weak, whose observation typically requires a resonant excitation. Second, the RBM mode is of very low frequency, which is out of the spectral range of our Raman setup.

In response, we have added more Raman spectra taken from a large number of samples into Fig. S3 of the Supplementary Information. Additionally, we have also cited Picheau's paper [ACS Nano 15, 596–603 (2021)] in the revised main text, as kindly suggested by the referee.

Figure R1. Raman spectra of more collapsed CNTs and Raman mapping. **a**, Raman spectra of more collapsed CNTs. **b**, Raman spectra of more free-standing CNTs. **c**, The distribution of FWHM of G peak of collapsed CNTs (red) and free-standing CNTs (black) in (a)(b). **d**, Topography image of both collapse and freestanding configurations, half underneath the hBN encapsulation and half expose in the outside. **e**, **f**, **g**, Near-field image, intensity of D peak mapping, FWHM of G peak mapping of the CNT sample shown in (a), respectively. Scale bar: 250 nm

[redacted]

Fig. R2. Uncertain change of Raman 2D peak intensity in the structural collapse. These two figures are taken from Fig. 1f and Fig. 5e of the paper Chen et al. [Nat. Electron. 4, 653–663 (2021)]. Panel a shows a slight increase of 2D peak after the structural collapse, whereas panel b shows a slight decrease of the 2D peak. Therefore, it is difficult to identify the collapsed CNT from the intensity of Raman 2D peak.

3. The observation (page 10, lines 202-203) that the same CNT can exhibit both circular and collapsed configurations under the same hBN flake is intriguing. The authors should provide further explanation and statistical data on the occurrence of such cases. Please explain why the collapse doesn't occur uniformly along length of the CNT. Raman mapping can be effectively employed to visualize how hBN encapsulation affects the Raman modes along the length of the CNT.

Our reply: We thank the referee for noticing the intriguing phenomenon that the same CNT can exhibit both the circular and collapsed configurations. This is likely caused by the uneven stress exerted on the CNTs by the hBN flakes on the silicon substrate. We noted that the CNT underneath experiences significant pressure only when there is localized tension strain within the hBN flake. However, the surface of the silicon oxide substrate is uneven, leading to varying tension or compression in different areas of the transferred hBN layers, resulting in non-uniform stress on the same hBN flake. Consequently, CNTs underneath the hBN flake undergo non-uniform pressure, allowing for the coexistence of both circular and collapsed configurations.

The occurrence of such phenomena depends on several factors. Longer CNTs are more likely to exhibit both circular and collapsed configurations due to the non-uniform distribution of stress. Additionally, the thickness of the hBN cover determines its rigidity and stiffness, which can influence the applied stress to the CNTs and the occurrence of such phenomena. Overall, the probability of such cases is more than 50% in our experiments.

We agree with the referee that Raman mapping provide a way to visualize how hBN encapsulation affects the Raman modes along the length of the CNT. Below, we show Raman 2D mapping of a CNT with both circular and collapsed configurations under the same hBN flake in Figure R3. Although one can identify the collapsed and circular sections from the Raman mapping, the spatial resolution of the mapping image is rather poor due to the diffraction limit of light (the laser spot size is about 800 nm). Besides, geometrical defect of the top hBN surface can also affect the Raman signal through the local field factor, inducing that the Raman mapping shape does not fully coincide with the CNT. In contrast, near-field optical technique can break the diffraction limit of light and achieve much better spatial resolution down to ~ 10 nm range, as shown in Fig. R3b and Fig. R10 below. Therefore, the near-field optical technique provides a superior method for visualizing the structural phase of CNTs underneath the hBN cover.

Figure R3. Raman mapping of both circular and collapsed sections of the same CNT. **a**, Topography image of both circular and collapsed configurations under the same hBN flake. **b**, **c**, **d**, Near-field infrared image, D-peak intensity mapping, G-peak FWHM mapping of the sample shown in panel (a), respectively. Scale bar: 250 nm

4. In general, it would be beneficial for the scientific community to understand how to distinguish between cylindrical and collapsed CNTs covered by hBN (or graphene) based on Raman spectra. Ideally, Raman maps of both free-standing and collapsed (under the hBN) regions of the same CNT should be measured to compare the frequency shifts and intensity variation in the D-band, G-band and RBMs, as a function of coordinates on the sample surface.

Our reply: We thank the referee for his/her constructive suggestion.

According to our Raman data and prior research, we can infer that there are at least four distinctions in the Raman spectra that can be used to differentiate between cylindrical and collapsed CNTs:

1. The Raman D-peak increases.
2. The Raman G-peak becomes significantly broader.
3. The Raman G-peak experiences a blue shift.
4. The Raman RBM mode disappears.

These differences in the Raman spectra are clearly depicted in Fig. R4 (also in Fig. 1e and 1f of the main text) and Fig. R5.

Figure R4. Raman spectra for both circular and collapsed CNTs. **a**, Raman spectrum of a round CNT (purple curve) and **b**, hBN-encapsulated collapsed CNT (green curve). The dashed lines indicate the initial G peak position of the round CNT. The red and blue lines are Lorentz fitting the D and hBN in-plane phonon peaks, respectively.

[redacted]

Figure R5. Disappear of RBM mode after the structural collapse. (Adapted from Nat. Electron. 4, 653–663 (2021))

5. Provide Raman spectra of pure hBN in areas free from CNTs to confirm that the peak at 1360 cm⁻¹ originates from hBN and not from a D-band component of the CNTs. It would be instructive to see how the pressure estimated from the shift of the hBN line aligns with the estimated pressure from the CNTs'

G-band shift. For reference, consider the frequency of few-layer (unstrained) hBN reported by Gorbachev et al. [Small 7, 465–468 (2011)].

Our reply: We appreciate the referee's helpful suggestion. Indeed, the differentiation between the hBN and D-band components from the CNTs can be verified by comparing the hBN area with and without CNTs. To address this, we conducted additional Raman measurements on pure hBN in CNT-free areas, as depicted in Figure R6 below. It is worth noting that hBN demonstrates a Raman peak centered around 1360 cm^{-1} , corresponding to the in-plane optical phonon mode of hBN. This differs from the Raman spectrum obtained from collapsed CNT beneath hBN, which exhibits a relatively broader Raman peak ranging from 1330 cm^{-1} to 1360 cm^{-1} . This broader Raman peak can be separated into two Raman modes: one representing the in-plane phonon mode of hBN at 1360 cm^{-1} , and the other being the D-mode of the CNT at 1330 cm^{-1} .

Regarding the shift of the hBN phonon mode, we would like to emphasize that the strain in hBN does not align with the strain in CNTs, as elaborated below. Firstly, only a small portion of hBN comes into contact with the CNTs, while the majority of hBN lies directly on the SiO_2 substrate. Consequently, the strain in hBN is primarily determined by its contact with SiO_2 . Secondly, the very small portion of hBN that directly sits on CNTs experiences predominantly tension strain rather than compression strain. This tension strain induces a red shift in the in-plane phonon mode of hBN. However, observing this red shift is challenging due to the nanometer-scale width of the hBN strain area (comparable to the diameter of CNTs) and the typically micrometer-scale Raman laser spot (due to the diffraction limit).

Figure. R6. Raman spectra of pure hBN. a, Raman spectra collected from different points of a hBN flake. **b,** Raman spectra of pure hBN and hBN-encapsulated CNT.

6. The almost linear trend observed for the CNTs in Figure 1d is rather puzzling. It appears as if the CNTs were manually selected for presentation and do not follow the normal distribution of diameters, as shown in Figure 3a (top panel). Please clarify how the CNTs in Figure 1d relate to those in Figure 3a. If only few CNTs were selected for Figure 1d, please provide a separate figure with the full statistics, for instance in the SI. Furthermore, it is not clear how many CNTs were studied in total in Figure 1 and Figure 3. Please specify the exact number in the text, rather than just providing a percentage.

Our reply: We thank the referee for pointing out this concern, which provide us a chance to improve the manuscript.

Although the roughly linear trend in Figure 1d seems to be discrepancy with the statistics in Fig. 3a, one should note that the diameter of 9 CNTs out of the total 15 CNTs ranges between 1.2nm and 1.6nm (see below in Fig. R7), which qualitatively match with Fig. 3a.

The remaining discrepancy comes from the different statistical methods. In Fig. 3a, the percentage refers to the probation of CNT length at a specific height. It is the length other than the number of CNTs that is counted here. As for many encapsulated CNTs, both circular and collapsed parts co-exist in the same individual CNT, such CNTs cannot be simply classified into either circular or collapsed CNTs. Therefore, we can only count the length of the circular and collapsed CNTs in Fig. 3a. The total length of CNTs for the statistics is 36,800 nm.

Following the referee's suggestion, we have added the relevant instructions in our revised manuscript, which reads as: "The percentage refers to the ratio of length between the collapsed CNTs and all CNTs studied."

In addition, we also provided the exact number of CNTs that counted in our study in the revised manuscript. There are 84 as-grown CNTs counted in the top panel of Fig. 3a and 15 hBN-encapsulated CNTs in the bottom panel. Besides, extra 18 CNTs encapsulated by graphite are counted in Fig. S5.

Figure R7. Statistics of the CNT height with (red squares) and without (black squares) hBN encapsulation. Noted that the diameter of 9 CNTs out of the total 15 CNTs ranges between 1.2 nm and 1.6 nm.

7. The authors repeatedly emphasize that they achieve local high pressures of up to 10 GPa. However, in the reviewer's opinion, their experimental data do not provide sufficient evidence to support such claims. The use of a convolution model, in this regard, is not a strong argument and raises further questions (see below). In fact, the average pressure estimated by the authors from the G-band shift of CNTs is only approximately 1.3 GPa. This estimation aligns closely with the findings of Vasu et al. [Nat. Commun. 2016], where a similar approach was employed to apply pressure to encapsulated molecules, resulting in a reported value of about 1.2 GPa. Furthermore, it appears that none of the works referenced in the review paper [He, Small 2019, 15, 1804473] report that the 10 GPa pressure is necessary for collapsing CNTs. Consequently, the assertion of reaching pressures of 10 GPa requires stronger justification and additional experimental data.

Our reply: We thank the referee for his/her comment.

Indeed, the maximum pressure of ~ 10 GPa is not a direct measurement result. Instead, it is estimated from the deconvolution of the Raman G-peak. Such a high pressure is also supported by our molecular dynamic simulations, as shown below in Fig. R8 (also in Fig. 2b of the main text). Noted that the pressure here generated by the vdW encapsulation is not uniform as the hydrostatic pressure. Instead, it varies strongly from place to place. The value of 10 GPa is the maximum pressure applied to the CNT, other than the average value of pressure.

To be more conservative, we have removed the “10 GPa” from the abstract, the introduction, the conclusion, and many other places of the manuscript.

Figure R8. Spatial distribution of the vdW pressure acting on a collapsed CNT.

8. In SI Section S2, the discussion of the deconvolution results appears to yield strange results, particularly the authors should clarify how they interpret negative pressure shown in Figure S3 and its implications on CNT collapse. Overall this section looks rather weak in terms of arguments. Consider revising it.

Our reply: We thank the referee for raising this interesting question.

The Raman G-peak shift and broadening can be explained by the strain in C-C bond. The tension/compression strain of the C-C bond led to the red-/blue-shift of the Raman G-peak. The negative pressure shown in Figure S3 reflects the existence of some slightly longer C-C bonds (tensile strain) in the collapsed CNT.

To avoid any confusion, we have replotted the figure using both the C-C bond strain and the van der Waals pressure as the x axis, as shown below in Fig. R9.

Figure R9. Deconvolution results of the Raman G peak of a collapsed CNT.

9. The manuscript presents only one example of the metallic-to-semiconducting transition in collapsed CNTs, which is statistically insignificant, especially when claiming it as a new method for obtaining collapsed CNTs with a tunable energy bandgap. Please provide information about the number of collapsed CNTs that exhibit a semiconductor behavior. Again, Raman spectroscopy provides a perfect tool for studying metallic-to-semiconductor (or semiconductor-to-metallic) transitions since it should be reflected in the presence or absence of Breit-Wigner-Fano profile in the G-band spectrum (see Okuno et al. [PRL 111 216101 2013]).

Our reply: We appreciate the referee's feedback.

In addition to the example presented in the main text, we have also observed the metallic to semiconducting transition in another 7 collapsed CNT samples, as displayed in Fig. R10 in below.

As noted by the referee, the Raman spectra of metallic and semiconducting CNTs differ in their Raman G-peak, specifically in the presence or absence of a Breit-Wigner-Fano profile. However, applying this method to hBN-encapsulated CNTs is challenging due to the strong distortion and broadening of the Raman G-peak in collapsed CNTs (approximately 3-5 times broader than the pristine G-peak).

In this study, we utilized scanning near-field optical microscopy (SNOM) to determine whether a CNT is metallic or semiconducting based on its response in the far infrared range, as shown in Fig. R10. A metallic CNT exhibits a strong response and can support Luttinger-liquid plasmons in the far infrared range due to the existence of free charge carriers. Conversely, a semiconducting CNT has a much weaker infrared response. This method allows for the straightforward identification of a CNT's metallic or semiconducting nature.

In response, we have included these images in the revised Supplementary Information.

Figure R10. More metallic CNTs that exhibits a semiconducting behavior after collapse. a-f, Topography image of the BN-encapsulated CNTs. **g-l,** Near-field infrared images of the tubes shown in a-f. The yellow arrows refer to the as-grown metallic CNTs, and the red arrows refer to the semiconducting part after collapse. Scale bars: 200 nm

10. In SI Section S3, it would be interesting to know if the authors measured Raman spectra for CNTs encapsulated in few-layer graphene and how the Raman lines of graphene and CNTs behave under strain. It will definitely have an added value for the community.

Our reply: We thank the referee for raising this interesting question. However, obtaining Raman spectra for CNTs encapsulated in few-layer graphene is highly complex. Firstly, distinguishing and isolating the Raman signals of CNTs from those of graphene is very challenging as they show almost the same Raman spectrum. Secondly, the Raman signal from CNTs is significantly smaller compared to that of graphene due to their reduced dimensionality. Thirdly, the graphene coverage will strongly screen the Raman signal from the underneath CNTs.

Minor Revisions:

- 1. On page 4, line 56, the references are incorrectly shown as "sample161920." Please correct the reference format.*
- 2. Replace "fermi" with "Fermi" throughout the text (e.g., page 10, lines 211 and 213).*
- 3. On page 13, line 266, change "Discussion" to "Conclusions."*
- 4. Please increase the resolution of the figures in the SI. Some parts of the figures, such as the pie chart in Figure S4b, are hard to see when zoomed in.*
- 5. Specify in the text how the error bars in Figure 1d (and other similar figures) were obtained.*

Our reply: We thank the referee for pointing out our mistakes. We have corrected all these minor issues in the revised manuscript.

Specifically, we have replotted the pie chart in the SI with higher resolution as shown below.

Figure R11. Revised figure S5 of the Supplementary Information section 5 - the height statistics of graphite-encapsulated CNTs.

Response to Referee #3

Our reply: We thank the referee for letting us know about his/her involvement in the Nature Communications initiative. It's commendable to see efforts towards training in peer review and recognizing the contributions of Early Career Researchers. The referee's commitment to this initiative is valuable for the research community.

Response to Reviewer #4

The manuscript by Hu et al (NCOMMS-23-40743) represents an interesting study of sandwiching carbon nanotubes between hBN flakes (or few layer graphene) and a silicon oxide surface. The vdWs attraction between the hBN and the SiO₂ presumably generates enough clamping force normal to the surface to fully collapse the nanotubes. Metallic nanotubes are apparently rendered semiconducting. Modeling complements the experimental studies.

I find the topic quite interesting-- it applies the now-routine but powerful methods of 2-D vdWs heterostructure engineering to the mechanical and electronic physics of 1-D nanostructures. If the authors can successfully address my concerns stated below, the paper could yield a nice Nature Communications publication.

Our reply: We thank the referee's appreciation of our work and his/her recommendation for publication in Nature Communications.

1. The paper makes no mention of the rich history of collapsed carbon nanotubes, which dates back nearly 30 years. The reader is led to believe that this is the first experimental observation of collapsed nanotubes! Either the authors are woefully ignorant of the extensive literature on this topic, or they have omitted important references in order to falsely elevate the importance of their work.

The introduction should be rewritten to include important past work in this area, such as the original discovery of collapsed carbon nanotubes (N. Chopra et al, Nature 377, 135 (1995)) and studies of the mechanical stability of inflated/collapsed nanotubes (Barzegar et al, Nano Letters 16, 6787 (2016); Magnin et al, Carbon 178, 552 (2021); Xiao et al, Nanotechnology 18, 395703 (2007)).

Our reply: We thank the referee for his/her constructive suggestion. We have added a new paragraph to the introduction part to review the history of collapse of CNTs, which reads as below:

"On the other hand, the collapse of CNTs has been discovered for a long time. Various methods have been reported for inducing the collapse of CNTs, such as applying hydrostatic pressure, using electron beam irradiation and employing ball milling. Recently, Chen et al. has demonstrated the ability to

collapse carbon nanotubes into nanoribbons less than 10 nm wide using high pressure generated by diamond anvils and thermal treatment. However, some methods are challenging to apply at the nano-micro scale, while others are complex and inefficient. Additionally, theoretical studies have presented a detailed collapse phase diagram of CNTs and identified the pressure range that CNTs with different diameters and walls could withstand without collapsing."

Additionally, we also included the references mentioned by the referee in the revised manuscript.

2. The theoretical studies/simulations presented in the paper assume a perfect single wall carbon nanotube (e.g. a 10, 10 tube). Yet the experimental part of the paper says almost nothing about the nanotubes used (other than that they were grown using CVD and what their apparent diameter is via AFM). CVD growth often yields highly defective tubes, with many kinks and "bamboo-like" internal structure. Such defects could completely alter the collapse energetics. How many walls do the nanotubes have? How defect free are they? If the tubes are multi wall, then the energetics are completely different, as only the inner-most walls contribute to vdWs energy lowering by collapse, whereas the outer walls only increase strain energy by collapse. Simply saying the thickness of the collapsed tubes as determined via AFM (through multiple layers of hBN or graphene) is that of bilayer graphene is not sufficient.

The authors must fully characterize the nanotubes employed, using high resolution TEM.

Our reply: We thank the referee for pointing out that we missed telling the structure of the CNTs used in this study, and for raising questions regarding the quality of the CNTs.

The CNTs used in this study are single-walled nanotubes grown using CVD method. The CVD growth of CNTs were conducted at a relatively lower temperature of 850°C, and with small-sized metal particles as catalysts. Such growing conditions typically generate single-walled CNTs with a diameter ranging from 1nm to 2nm. This information of single-walled structure has been added into the revised manuscript, which reads as *"The single-walled CNTs were initially grown on SiO₂ substrate through chemical vapor deposition (CVD), and then covered by an hBN flake through mechanical transfer."*

Regarding the quality of the CNTs, while there were previously issues with producing defective tubes, such as "bamboo-like" tubes, recent advancements in the CVD growth have led to the mature growth of high-quality single-walled CNTs. For instance, the CVD growth has produced the highest-quality single-walled CNTs that exhibit quantum conductance and Fabry-Pérot interference [*Nature* 424,654(2003); *Nature* 411, 665 (2001)].

Our study follows similar growth methods as in previous studies, and we have used AFM, Rayleigh scattering spectroscopy, and scanning near-field optical microscopy (SNOM) to characterize the diameter and optical response of the CNTs, as shown in Fig. R12 below. The results demonstrate that the CNTs used in this study are single-walled and of high quality, exhibiting excellent optical properties [*Nature Nanotechnology*, 7, 325–329 (2012)]. The near-field infrared image presents the interference fringes from the interference of plasmon polaritons in CNTs, confirming that the CNT is single-walled and of high quality [*Nature Photonics*, 9, 515–519 (2015); *Nature Materials*, 19, 986–991 (2020)].

[redacted]

Figure R12. As-grown high-quality single-walled CNTs. **a**, Topography image (left, gray) and near-field infrared image (right, colored) of an as-grown CNT. **b**, Rayleigh scattering spectrum of an as-grown CNT with chiral index (21,7). (adapted from our previous literature *Appl. Phys. Lett.* 117, 023101 (2020))

3. Several aspects of data presented in Fig. 1 don't make sense to me. In Fig. 1c, the FWHM of the topography profiles are identical for the inflated and collapsed tubes. How can this be? The hBN surely additionally smears out the line profile, and the collapsed tube is intrinsically wider than the inflated tube.

Also, for many (most) tube parameters, the schematic of Fig. 1a is unrealistic: The flattened portion of the tube significantly lowers the activation barrier or collapse, so the "exposed" part of the tube should spontaneously collapse as well. Do substrate interactions somehow keep it from zipping closed?

How are Raman spectra of Figs. 1e, 1f collected? Is the signal really from just one tube? Is some sort of near-field optical method used? I find it hard to believe that the Raman signal is collected from just the very small circular spots on a single tube as shown in Fig. 1b.

Our reply: We thank the referee for bringing these concerns.

- a. As noted by the referee, the FWHM of the topography profiles for circular and collapsed nanotube are indeed roughly of the same value of ~30nm. However, these measurements do not accurately represent the actual width of the inflated or collapsed tubes (in the range of 1 nm to 3 nm). The largely increased FWHM is a result of the size effect of the AFM tip, which typically has a radius of 10-30 nm. Images obtained by scanning a sample with an AFM tip are strongly dependent on the AFM tip geometry. AFM scans are a result of the convolution of the AFM tip and the surface. The smaller the surface features compared to the AFM tip dimensions, the less accurate the sample image on the computer screen. Conversely, the measured height value in the z-direction is not affected by the tip size effect and can be considered as the true diameter of carbon nanotubes.
- b. Regarding the spontaneous collapse, for a single-walled CNT without external pressure, the critical diameter for spontaneous collapse is approximately 5 nm (*Carbon* 2021, 178, 552-562). This means that for a single-walled CNT with diameter smaller than 5 nm, the formation energy of the circular state is lower than that of the collapsed state. Conversely, for a single-walled CNT with diameter greater than 5 nm, its formation energy of the circular state is higher than that of the collapsed state. The critical collapse pressure as a function of the tube diameter calculated by ourselves is presented below in Fig. R13. The diameters of our experimental CNTs are about 1~2

nm, which is much smaller than the critical diameter of 5 nm. As a result, the "exposed" part of CNT will not collapse spontaneously.

- c. The Raman spectra of Figs. 1e, 1f were collected from the same tube shown in Fig. 1b, using far-field Raman spectroscopy technique. The actual spot size (or spatial resolution) is about $1\ \mu\text{m}$. The circular marks denote the location from where we collect the Raman signal, other than the real Raman spot size.

Figure R13. Critical Collapse pressure as a function of the tube diameter.

4. The paper suggests a metal to semiconductor transition induced by the hBN coverage and collapse. But of course not all SWNTs are metallic to begin with. What happens if the tube is intrinsically semiconducting? Does a semiconducting to metal transition take place? As Lammert et al (*Phys. Rev. Lett.* 84, 2453 (2000), ref. 37 in the manuscript) show, tube collapse causes metallic (n,n) tubes to become semiconducting and can also cause small-gap semiconducting tubes to become metallic. Why is this not observed by the present authors?

Our reply: We thank the referee for his/her thoughtful comment.

We fully agree with the referee that tube collapse can also cause small-gap semiconducting tubes to become metallic, as predicted by Lammert et al [*Phys. Rev. Lett.* 84, 2453 (2000)].

We have also performed first-principles calculations on semiconducting SWCNTs with a specific chirality. Our results show that a semiconductor-metal transition would occur in SWCNT under sufficient pressure, as shown in Fig. R14 below, consistent with previous findings [Lammert et al, *Phys. Rev. Lett.* 84, 2453 (2000)]. However, the estimated pressure required to induce such a transition in a semiconducting (11, 0) zigzag SWCNT is approximately 6 GPa, which is higher than the average vdW pressure that can be achieved. Semiconducting SWCNTs with larger band gap require even higher pressure to induce a semiconductor-metal transition. To experimentally observe the semiconductor-metal transition in hBN-encapsulated CNT, the CNT requires to have an appropriate band gap in a small energy range. On the one hand, the band gap should not be too large, as it would exceed the vdW pressure that hBN can apply. On the other hand, the band gap should not be too small, as the CNT with small band gap behave like metallic at room temperature due to the thermal excitation [such as (3N, 0) zigzag CNT]. These constraints strongly impede the experimental observation of the semiconductor-metal transition in CNTs.

Conversely, for the metallic SWCNT, small pressure can induce a metal-semiconductor transition. Therefore, it is easier to observe such a transition experimentally. Additionally, the encapsulation by hBN flakes will unavoidably decrease the infrared response of the underneath CNTs, which may submerge the observation of the enhanced infrared response of the collapsed CNTs, making it difficult to observe a transition from initially semiconducting to encapsulated metallic experimentally.

[redacted]

Figure R14. Structural change induced semiconductor-metal transition in a (11, 0) zigzag single-walled CNT. A high pressure of ~6 GPa is required for the semiconductor-metal transition.

5. The Supplementary Information has some confusing parts. For example, section S1 is entitled "The simulation of the influence of structural transition of the hBN", yet the text there (and Fig. S1?) deals with graphene, not hBN.

Our reply:

We thank the referee for pointing out our mistake. We have corrected it in the revised version of the Supplementary Information.

Response to Referee #5

In the manuscript titled "Collapse of carbon nanotubes by local high-pressure from van der Waals encapsulation" Hu, Shi and colleagues report an interesting phenomenon where 1D carbon nanotubes (CNTs) are collapsed by encapsulating them in between a 2D graphene crystal and SiO₂ surface. The main finding of the manuscript is that carbon nanotubes, regardless of their initial diameter, when placed in between a hBN flake and SiO₂ surface show a uniform collapsed thickness of 0.7 nm. They confirmed the collapsed structure mainly by AFM and Raman spectroscopy. Overall, the manuscript is well written and has very detailed theoretical simulations.

While I believe this is an important work and can have interesting applications as 1D metal semiconductor junctions, the manuscript lacks detailed characterisation and convincing experiments.

Our reply: We thank the referee for his/her very positive evaluation of our work. Below we address the questions and comments raised by the referee one by one.

1. The authors have stated that a high proportion (77%) of the CNTs flatten under the top hBN layers. Is this constant across all the samples measured? Since the deformation of the CNTs depend on the force applied on individual tubes, I would expect that if there is sufficient number of CNTs trapped in between hBN and SiO₂ layers, the tubes would avoid collapsing under the vdW pressure. Have the authors noticed any dependence on the number of CNTs trapped underneath the hBN layer and the percentage of collapsed tubes?

Our reply: We thank the referee for raising questions regarding the dependence of the collapsing ratio on the CNT density.

We agree with the referee that the collapsing ratio could be lower for dense CNTs trapped underneath the hBN. If the CNT density/number is high enough, the tubes would avoid collapsing. However, we haven't observed such dependence because the density of CNTs used in our experiments is extremely low, typically 3 μm apart from each other, as shown in Fig. R15 below. We believe that the collapsing ratio can only be significantly affected when the density of the CNT network is so high that the distance of the adjacent interval is less than the distance of hBN stress relaxation length, which is in the order of tens of nanometers based on our molecular dynamics results.

In addition to the CNT density, we believe that the collapsing proportion should also be affected by other factors, such as hBN thickness, nanotube diameter, and bottom substrate surface roughness.

Figure R15. Near-field infrared images of typical as-grown CNTs. a, b, The typical distribution of as-grown CNTs on the SiO₂ substrate. The density of the CNTs is rather low, and they are typically spaced micrometers away from each other.

2. The authors should provide a larger area image of the encapsulated sample to give readers a better idea if there are bundles or individual nanotubes of encapsulated CNTs

Our reply: We thank the referee for this constructive suggestion.

Following the referee's suggestion, we provide large-area images of CNTs, as shown in below Fig. R16, with arrows referring to the hBN-encapsulated CNTs. Noted that these images are near-field infrared images other than topographic images, as the small diameter of the CNTs (1-2 nm) makes them difficult to discern in large area topographic images. These large-area near-field infrared images reveal that the nanotube density is quite low, typically spaced a few micrometers apart from each other. This low density is attributed to the extremely sparse distribution of the catalytic Fe nanoparticles used in the CVD growth process. Consequently, the resulting nanotubes are predominantly individual, with few forming bundles.

We have added the following images into the revised Supplementary Information.

Figure R16. The near-field infrared images of typical distribution of hBN-encapsulated CNTs. Red arrows refer to the hBN-encapsulated CNTs.

3. I think the experimental section would benefit from more details of sample preparation especially how the hBN is placed on top of the CNTs. On a similar note, it is not mentioned if the Fe nanoparticle capping and the end encapsulation were removed before the transfer of hBN sheets.

Our reply: We thank the referee for the constructive suggestions.

The hBN is transferred onto CNTs using a dry transfer method commonly used in research [*Nature Communications* 2016, 7, (1), 11894; *npj 2D Materials and Applications* 2019, 3, (1), 22]. The specific process involves exfoliating hBN flakes onto a polypropylene carbonate (PPC) coated polydimethylsiloxane (PDMS) block to create a stamp. The hBN is then aligned on the stamp with the targeted CNT and released. Upon heating the substrate to over 120 °C, the hBN sheet adheres to the CNT after the stamp is lifted, as the adhesiveness of the PPC decreases.

In our experiments, the Fe nanoparticles were not removed before the hBN transfer. As a result, the nanoparticles remained at one end of these nanotubes throughout the duration of the experiments.

Figure R17. Dry transfer process of hBN flakes.

4. Figure 1d raises quite a few questions. a) What is the red band in the figure stand for? b) do the sample index denote the number of nanotubes measured? In that case are there only 15 nanotubes trapped under the hBN flake? and c) how do the authors achieve the error bars? are they from measurements from the same nanotube at different positions or from different nanotubes of similar diameter?

Our reply: We thank the referee for raising these questions.

- The center of the red band represents the theoretically calculated height of the collapsed carbon nanotubes. This information has been added to the figure caption of Fig. 1d in the revised manuscript.
- In our study, we have identified much more than 15 nanotubes that are trapped beneath the hBN flakes. However, in Figure 1d, we only count the CNTs that are partially enclosed in very thin hBN flakes (less than 5 nm thick) and partially exposed, for two reasons. Firstly, we are only able to measure the diameter of the CNTs and the height after their collapse for those samples were partially collapsed and partially uncovered by hBN. Secondly, the accuracy of the height measured

by AFM is affected by the thickness of the hBN (typically requires very thin hBN flakes to achieve accurate measurement).

- c. The error bars represent the standard deviation of multiple height values measured at different positions of the same carbon nanotube. This information has also been added to the figure caption of Fig. 1d.

5. In Figure 1e and f are the spectra collected from a single nanotube? Did the authors look at the RBM peak position of the collapsed and round nanotubes? It might give interesting information such as pressure and nanotube width. Moreover, the G-band broadening should also be analysed by deconvolution. The authors have carried out some deconvolution studies in figure S3- which is very difficult to understand and does not have any reference if any reported protocol was followed.

Our reply: We thank the referee for raising these questions.

The Raman spectra in Figs. 1e and 1f are collected from the same nanotube. From the two Raman spectra, one can clearly see that the CNT Raman G-peak got shifted and broadened after the structural collapse. Additionally, D-peak occurs after the collapse, reflecting the imperfection of the structure.

We thank the referee for his/her constructive suggestion on looking at the RBM mode. However, it is difficult for us to measure the RBM peak. First, the RBM peak is very weak, whose observation typically requires a resonant excitation. Second, the RBM mode is of very low frequency, which is out of the spectral range of our Raman setup. As the Raman G-peak shift and broaden have already demonstrated clearly the structural collapse, we haven't devoted significant effort on measuring the RBM mode.

The broadening of the Raman G-band is attributed to the uneven distribution of strain in the collapsed CNTs. Basically, the tension/compression strain in the C-C bond will lead to the red-/blue-shift of the Raman G-peak. The combined effect of these shifts results in the broadening of the Raman G-band. As mentioned by the referee, we have performed deconvolution analysis of the broadened G-band using the original Raman spectrum taken from Fig. 1f. The relationship between Raman G-band shift and external pressure (strain) is based on the findings in the literature [Nature Electronics 2021, 4 (9), 653-663]. Further details of the deconvolution method employed can be found in chapter 6 (Convolution, Correlation, and Power Spectral Density) of the book [Gan, W.S. (2020). *Signal Processing and Image Processing for Acoustical Imaging*. Springer, Singapore].

6. Have the authors considered the errors associated with AFM measurement of nanotube height as the nanotubes can deform under pressure from AFM tip itself? (Nanomaterials 2023, 13(3), 477)

Our reply: We thank the referee for raising this interesting question.

The scanning mode used in [Nanomaterials 2023, 13(3), 477] is peak force mode. In this mode, the AFM tip works in the repulsive interaction region and applies a moderate compress to the sample. As a result, the nanotube height in the literature is deformed by the AFM tip, leading to errors in measured diameters.

In contrast, our study uses tapping-mode AFM for scanning the CNTs, keeping the AFM tip in the attractive interaction range. This approach avoids applying compression to the CNTs and provides measured values representing the real diameter of CNTs [Precision Engineering 2020, 64, 269-279].

Tapping-mode is the most commonly used mode in AFM, where the cantilever oscillates at its resonance frequency near the sample surface. Feedback electronics are used to control the distance between the cantilever and the sample by maintaining a constant amplitude or phase of the oscillation. This technique avoids lateral tip-sample forces, leading to higher resolution compared to the traditional contact-mode AFM. Consequently, tapping-mode AFM is extensively employed for accurately measuring the size of nanostructures with great precision and reliability.

7. The characterization of CNTs is incomplete. Are the CNTs synthesized single or multi walled? Is there any dependence of buckling pressure on the metallicity or chirality of the CNT? In their report R.S. Alencar et al. (Carbon, 2017, 125, 429-436) theoretically predicted that collapse pressure depends on the diameter of the nanotubes. Do the authors see any such dependence?

Our reply: We thank the referee for pointing out that we missed telling the structure of the CNTs used in this study. The CNTs used in this study are single-walled nanotubes grown using CVD method as that reported in previous literatures [*Nano Letters* 2019, 19, (4), 2360-2365; *Nature Materials* 2020, 19, (9), 986-991]. This information has been added into the revised manuscript, which reads as "*The single-walled CNTs were initially grown on SiO₂ substrate through chemical vapor deposition (CVD), and then covered by an hBN flake through mechanical transfer.*"

As expected by the referee, the collapse pressure of nanotubes is indeed related to their chirality. In most of the chiral cases, the collapse pressure is very close, but in the case of two special chiralities, namely armchair or zigzag, the collapse pressure will decrease significantly. The difference in the probability of collapse due to the chirality of carbon nanotubes is difficult to study in our experiment. This is because the stress applied by the hBN to the CNTs is neither uniform nor controllable. Therefore, it is difficult to get quantitative conclusions by using this method.

The purpose of this study is to demonstrate that the van der Waals encapsulation can trigger structural collapse of CNTs. Studying the dependence of the critical pressure for collapse is out of the scope of this study and requires a method that can provide quantitative control of the applied pressure on the CNTs.

8. In Figure 2a, authors state that "The red arrows represent the direction of the force on the atoms, and the lengths represent the magnitude." The magnitude of force is impossible to understand without a scale bar.

Our reply: We thank the referee for his/her suggestion. We have inserted a scale bar in the revised Fig. 2a of the manuscript. The revised figure is also attached below as Fig. R18.

Figure R18. Molecular dynamics (MD) simulations of the hBN-encapsulated CNTs. **a**, Sectional view of a hBN-encapsulated CNT simulated by molecular dynamics. The red arrows represent the direction of the force on the atoms, and the lengths represent the magnitude. The length of the arrow in the upper right corner of the plot represents 200 pN. **b**, Spatial distribution of the vdW pressure/force acting on the CNT. **c**, Extracted height of the round CNT (black) and the hBN-encapsulated collapsed CNT (red). CNTs with diameter from 0.8 nm to 2.0 nm collapsed under the vdW pressure to around 0.7 nm.

9. As far as I understand, the authors only perform AFM measurements to characterize nanotube dimensions. I do not understand how they have information for both height and diameter in figure 2c. Especially for collapsed nanotubes, I would expect their diameters to be higher compared to non-collapsed nanotubes. But that does not seem to be the case.

Our reply: We thank the referee for raising this concern.

For a circular CNT, its diameter equals to the height value measured by AFM. For collapsed CNTs, the diameter and height can only be measured simultaneously for samples that were partially collapsed by thin hBN flakes encapsulation and partially exposed. For those samples, the two values are achieved at different parts of the same CNT. The diameter of CNTs is determined by measuring the height of the circular part without hBN encapsulation, and the height is measured at the part with hBN encapsulation. Due to the structural collapse of CNT, the height decreases, and is typically smaller than the initial diameter.

We agree with the referee that it is likely easier to collapse for nanotubes with larger diameter. However, investigating the relationship between the critical collapsing pressure and tube diameter is beyond the scope of this study and necessarily requires a method capable of quantitatively controlling the applied pressure on the CNTs.

10. In figure 3, there is no information about how many nanotubes are considered for the statistical analysis. Moreover, larger nanotubes (1.8 nm and 2.0 nm) seem to be not as much affected by the vdW pressure. Is this assumption correct? If so, what are the possible reasons for such unusual behaviour?

Our reply: We thank the referee for pointing out the missing information about the number of nanotubes are considered for the statistical analysis. The statistical analysis was performed on 84 as-grown CNTs counted for the top panel of Fig. 3a and on 15 hBN-encapsulated CNTs for the bottom panel. This information has been added into the revised manuscript. It is important to clarify that the statistical findings are derived from the length of the nanotubes encapsulated by the hBN, rather than the quantity of nanotubes.

We particularly thank the referee for pointing out the unusual behavior that the larger-diameter nanotubes (1.8 nm and 2.0 nm) seem to be less affected by the vdW pressure. There are several potential explanations for this unexpected result: 1) the larger-diameter samples may possibly be bundles of CNTs, although the occurrence of such cases is rather rare; 2) the sample size may not be sufficiently large to draw definitive conclusions.

11. The theoretical simulation for van der Waals pressure assumes a flat top and bottom surface. But experimentally, the authors have used SiO₂ as bottom substrate, which is not atomically flat. Would there be any difference in vdW pressure for an uneven substrate compared to a flat substrate?

Our reply: We thank the referee for raising this question.

The rough SiO₂ substrate could prevent the hBN layer from sliding easily, causing strain in the hBN layer to be unevenly distributed. This uneven distribution of strain could lead to a nonuniform pressure on the encapsulated CNTs. As a result, we experimentally observed that many of the encapsulated CNTs are only partially collapsed. If the substrate were atomically flat, we would anticipate a uniform vdW pressure across the surface.

In our simulation, the periodic condition was used, which means that the top hBN is pinned by the boundary. This mimics the pinning effect from the rough SiO₂ surface.

12. Experimental details for scanning near-field optical microscopy sample preparation are lacking. I think the authors should add information about How the sample was prepared and the electrical contacts formed.

Our reply: We thank the referee for his/her suggestion.

The sample preparation and electrical contact fabrication are described in details below:

“Catalytic nanoparticles (Fe) were deposited on the SiO₂/Si chips through thermal evaporation (evaporation rate: 0.004 nm s⁻¹, base vacuum pressure: 1 × 10⁻⁶ mbar). Then the chips were put into a tube furnace (Anhui BEQ Equipment Technology), and gradually heated up to the CNT growth temperature (850 °C) under hydrogen and argon gas mixture at atmospheric pressure. When growth temperature was reached, argon was replaced by methane to commence CNT growth. After a growth period of 60 mins, the systems were cooled down to room temperature under a protective hydrogen and argon atmosphere.^{41, 42} After that, mechanically exfoliated hBN flakes on PPC film were transferred to the as grown CNTs on SiO₂/Si chips. At last, the encapsulated CNT samples were exposed to

hydrogen plasma at 280 °C to remove all organic residuals and contaminations. Standard electron beam lithography, electron beam deposition and lift-off technology were conducted to locate the Au electrodes on the freestanding part of CNTs.”

The above experimental details have been included in the Methods Section of the revised manuscript.

13. Since such semi-collapsed structure of CNTs would theoretically form 1D metal-semiconductor junctions, their electrical measurements should be very interesting. I would encourage the authors to include such kind of experiments in the manuscript.

Our reply: We thank the referee for this wonderful suggestion. Following the referee’s suggestion, we have measured electrical transport in CNTs with 1D metal-semiconductor junctions.

In order to achieve a more controllable metal-semiconductor junction, we employ an AFM tip to apply a local compress. Fig. R19a is a schematic diagram of our device. We first prepared a CNT fully encapsulated in hBN with two metal electrodes for the electrical measurement. Fig. R19b shows its resistance as a function of gate voltage (U_g), demonstrating its quasi-metallic nature. Subsequently, we adjusted the gate voltage to 2V, the source-drain voltage to 1V, and used the AFM tip in nap mode to apply varying pressures to the CNT, leading to changes in its electrical properties over time, as depicted in Figs. R19 c-f.

We found that when the tip applied pressure to the CNT, there was a significant increase in the resistance of the CNT, with the resistance increasing further as the pressure intensified, demonstrating the metallic-semiconducting transition in structural collapse. Interestingly, when the tip was subjected to the same pressure (dropped by 1000 nm), we observed that the resistance changed in response to the variation in gate voltage. In Figs. R19 g-j, we present the resistance of a CNT over time at different gate voltages (ranging from 0 V to 6 V). When the gate voltage alters the Fermi surface near the Dirac point of the CNT, the electrical properties are minimally affected by external pressure. However, at gate voltages (2V, 4V, and 6V) further from the Dirac point, there is a more pronounced change in resistance, which we attribute to the pressure increasing the bandgap and consequently elevating the resistance.

The above electrical measurements provide more evidence for the pressure-induced metal-semiconductor transition.

Fig. R19. Electrical transport measurement on a CNT with 1D metal-semiconductor junctions induced by an AFM tip.

Minor points:

1. The information about how the error bars are obtained and what they signify should be added to all the possible figures.
2. There is no Figure S2
3. In figure S4b, what does Length percentage signify?
4. Supplementary figures should be referenced as appropriate in the main text.

Our reply: We thank the referee pointing out our mistakes. We have made proper modifications in the revised version.

1. The error bars represent the standard deviation of multiple height values measured at different positions of the same carbon nanotube. This information has been included in the revised manuscript.
2. We have added Figure S2 in the revised Supplementary Information.
3. The length percentage refers to percentage of length. The statistical results are derived from the length of the nanotubes, rather than the quantity of nanotubes. We are counting the proportion of length in the CNT with different heights in Figure S5b (Figure S4b in original version). The length is counted here rather than the number of individual stems because, as mentioned earlier, there

are two different structural states of CNTs existing simultaneously in the hBN encapsulated region, and the length count is more meaningful. The same statistical method is used in Figure 3.

4. All sections of the Supplementary Information are referenced in the revised main text.

REVIEWERS' COMMENTS

Reviewer #1 (Remarks to the Author):

Reviewer #2 (Remarks to the Author):

I think the authors have well address all review comments. I suggest to publish as it is.

Reviewer #3 (Remarks to the Author):

Reviewer #4 (Remarks to the Author):

I commend the authors for taking seriously the comments of the referees. The revised manuscript is much improved. My original concerns have been adequately addressed, and I now recommend the paper for publication in Nature Communications.

Reviewer #5 (Remarks to the Author):

The authors addressed all the comments, and I recommend publishing it in Nature Communications.